# Tribological Examination of Anodized Al-356 for Automotive Use

Alexandra Musza [1,2], Márk Windisch [1,2], Mátyás Török [1], Tamás Molnár [1], Sándor Kovács [1], Szonja Sátán-Papp [1], Krisztián Szegedi [1], Dávid Ugi [2,3], Nguyen Quang Chinh [2] and Ádám Vida [1,*]

1   Bay Zoltán Nonprofit Ltd. for Applied Research, Kondorfa Utca 1, H1116 Budapest, Hungary; tamas.molnar@bayzoltan.hu (T.M.); sandor.kovacs@bayzoltan.hu (S.K.)
2   Department of Materials Physics, Eötvös University, Pázmány Péter Sétány 1/A, H1117 Budapest, Hungary
3   Research Centre for Natural Sciences, Institute of Materials and Environmental Chemistry, H1519 Budapest, Hungary
*   Correspondence: adam.vida@bayzoltan.hu; Tel.: +36-70-515-7182

**Abstract:** The A356 alloy is commonly used in the foundry industry to produce high-stressed automotive components, such as motor frames and cylinder heads. The aim of this work was to investigate how the mechanical and tribological properties of this alloy can be improved by applying an anodizing process. The properties of the oxide layer formed by anodizing using oxalic acid at low temperatures were characterized by different analytical and tribological methods. The combination of tribological methods with standard measurement techniques—such as hardness measurement, layer thickness measurement, as well as the analysis of the morphological characteristics—was used to track the layer evolution during wear developments.

**Keywords:** anodizing; automotive; tribology; A356 alloy

## 1. Introduction

Due to the dynamic development of the automotive industry, increasing demands arise, such as greater safety and reliability and further reduction of weight and fuel consumption (and thus emission), which cannot be met by "traditional" materials (such as steels) in all cases. As a result, there is an increasing need for alloys with lower density but adequate mechanical properties, including, e.g., high fatigue limit, high strength, and well-accepted plasticity [1,2].

Considering the modern material development trends, two directions can be observed. In the first trend, a significant proportion of research focuses on the development of high and ultrahigh-strength steels, while in the second one, the demand for the development of light metals, including aluminum, magnesium [3,4], and titanium alloys, is continuously growing. It is well known that titanium offers performance and mass-saving benefits in automotive components subjected to reciprocating and suspension loads and in those exposed to extreme temperatures and gradients. In this case, however, the raw material is expensive, and any special handling may prevent it from competing with steel in the most possible applications [5,6].

Aluminum is the third most abundant material available in nature. It has replaced the various types of steel in a wide range of applications due to having a strength-to-weight ratio superior to steel. Under the influence of oxygen, aluminum forms a natural protective oxide coating, which is why it is highly corrosion resistant. Various surface treatment methods, such as anodizing, painting, or varnishing, can further improve this property. Aluminum has a low melting point and density. In the molten state, it can be processed in several ways. Furthermore, it can be easily shaped, welded, and as metals usually are, it is recyclable, and its quality does not decrease significantly during the process. Remelting aluminum requires relatively low energy, as only about 5% of the energy required to

produce the primary metal is required in the recycling process. Although pure aluminum has excellent corrosion resistance due to the spontaneous formation of a passive aluminum oxide layer on its surface, it is rarely used in industries because of its relatively poor mechanical properties. By alloying elements, the mechanical properties can be improved; however, the corrosion resistance of Al alloys will decrease [7–9] in most of the cases. In the study of Sharma et al. [10], the properties of an aluminum-based composite were improved due to the presence of minerals reinforcement.

Surface modification with laser alloying can be also used to improve the properties of aluminum alloys. In the work of Staia et al. [11], a 2000-W-Nd-YAG laser was used to alloy a powder composed of 96 wt% WC, 2 wt% Ti, and 2 wt% Mg at different traverse velocities of 100, 200, 300, and 400 cm min$^{-1}$. Considering the roughness, morphological and microstructural characteristics, experimental results have shown that the wear resistance of the laser-treated surface is better than that of the substrate in all cases.

Agbeleye et al. were successful in adding clay to aluminum (AL6063) as a relatively cheap alternative, then studied the mechanical and tribological properties of these composites as brake pads [12].

Another way to increase the native oxide layer on the surface of aluminum alloys is electrochemical treatment, for example, anodizing [13]. The thickness, hardness, and wear resistance values of an anodic oxide layer on aluminum must be high enough for industrial purposes [14]. Anodizing, as a simple electrochemical method, converts aluminum into its oxide by appropriate selection of the electrolyte (usually an acidic solution) and the anodizing conditions, such as current density, voltage, and temperature. The process increases the thickness of the natural oxide layer [7].

Bensalah et al. investigated the effect of sulphuric acid anodizing conditions on the growth rate and density of anodic oxide layers with the Doehlert experimental design [15]. Tawakkal and Korda anodized 7075 T6 aluminum alloy specimens in tartaric and sulphuric acid (TSA) and subsequently protected them with a boiled water sealing treatment [16]. The average layer thickness was between 1.5 and 4.7 μm. Shao et al. used hard anodization (HA) and micro-arc oxidation (MAO) to prepare coatings with $27 \pm 3$ μm thickness onto the surface of 7050 Al alloy. The MAO coating, mainly composed of $\gamma$-Al$_2$O$_3$, was much more effective in isolating the substrate from the corrosive environment than the HA one due to the amorphous composition and the penetrating defects in the HA coating [17]. E. Santecchia et al. made an approximately 100 μm thick surface onto the aluminum alloy 6082 T6 with hard anodizing [18].

The hard anodizing process is practiced more and more often nowadays. In many applications, hard anodic coatings are competitors for hard chromium coatings. The former, when compared to the latter, are cheaper to apply, give better oil retention, and finally, their production is more environmentally friendly. However, they cannot be used on every type of aluminum alloy [19].

Hardness and wear resistance, or the better tunability of these properties, play a key role in the use of aluminum alloys. They can vary widely with the anodizing conditions, like temperature, electrolyte concentration, time, and current. The galvanostatic anodizing can be performed at current densities from 1 to 8 A/dm$^2$ in a mixed sulphuric acid–oxalic acid electrolyte at temperatures varied between $-5$ °C and $+20$ °C. It was found that in the temperature range between $-5$ and $+5$ °C, a constant wear resistance is present regardless of the applied current density [20].

The currently available literature mostly deals only with the mechanical properties of the anodized layer (hardness, roughness, wear property) [21–25].

The examination of this adhesion is important because, during automotive applications, the anodized layer can be damaged (wear, scratches, chipping, detachment, etc.), thus deteriorating the properties and resistance of the aluminum alloy.

Several models predict the formation of the anodized layer. The anodization of aluminum can be explained according to the Keller-Hunter-Robinson model [26,27], that the liberated oxygen reacts with the aluminum and turns into a dense, homogeneously

distributed oxide film, and from this, an oxide layer with a thickness dependent on the cell voltage is formed. The oxide film is dielectric in nature, so the current passes through it only if the metal ions leaving the base metal are able to diffuse in the oxide-electrolyte boundary phase. The formed aluminum oxide can be dissolved with the electrolyte, then the oxide layer loosens at the defect sites, and pores are formed, where the aluminum oxide is reformed. This process continues until the entire sample is covered with a porous layer, the thickness of which depends on the treatment time. The pore channels penetrate to the oxide barrier layer.

According to the Murphy-Michelson model [28,29], after the formation of the solid oxide barrier layer composed of $Al_2O_3$, a part of the aluminum oxide on the outer surface of the layer is transformed into hydroxide by water absorption, and the formation of the barrier layer is continuously renewed on the metal side. According to this theory, pore formation is not a prerequisite for oxide formation. During anodic oxidation, the oxide layer is not detached from the electrolyte onto the surface of the carrier but is formed through the transformation of the outer layer. It follows the physical and mechanical properties of the surface metal layer, its chemical composition, and its morphology.

Based on the nucleation model [30,31], the oxide formation does not start homogeneously on the metal surface, but tiny solution seeds appear scattered at given points. The process takes place so quickly that a homogeneous, continuous oxide layer can form. These nuclei depend on experimental parameters such as electrolyte concentration and cell voltage. The aim of Zhu et al. was to determine the influence of the Si content and the morphology of the Si particles on the anodizing reaction. Based on their results, by increasing the Si content, the Al phase appeared to be refined due to more nucleation of eutectic Si, and the fraction of the eutectic phase increased. In the unmodified alloys, the eutectic Si particles showed a polygonal flake shape and formed a continuous branched network [32]. This is also confirmed by our own measurement, that after anodization, an amorphous aluminum oxide layer is formed, with Si precipitates in it [33].

The aim of the present study is to develop a wear-resistant coating for automotive brake disks made of recycled aluminum alloy. The question was pointed: how do the surface and hardness change depending on varying the anodizing parameters, and with which of them can we achieve a good result? To find the best-performing surfaces, ball-on-disc tribological examinations were performed to further characterize the anodized layers.

## 2. Materials and Methods

### 2.1. Sample Preparation

The A356 alloy substrate was prepared from recycled aluminum waste (Figure 1). The nominal chemical composition of the untreated sample (A356 Al-alloy) was Al-7Si-0.29 Mg in wt%. This was checked also by EDS measurement; the results can be found in Table 1).

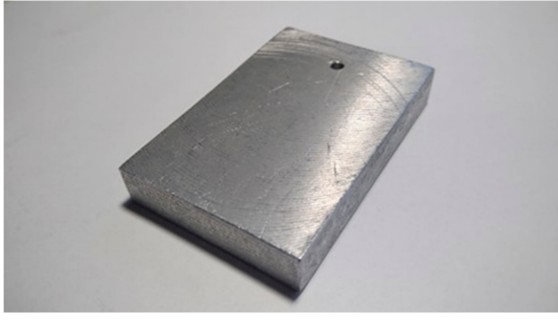 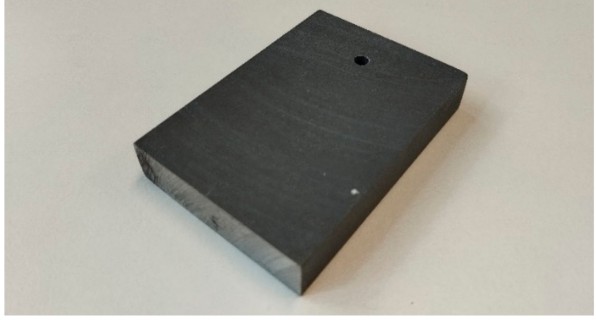

**Figure 1.** The untreated sample (**left**) and the anodized sample ((**right**), j = 2 A/dm$^2$ and t = 1 h).

**Table 1.** Chemical composition of A356.

|  | Al (wt%) | Si (wt%) | Mg (wt%) | Others (wt%) |
|---|---|---|---|---|
| literary | 92 | 7 | 0.35 | 0.65 |
| measured value | 91 | 8 | 0.35 | 0.65 |

To remove the natural oxide layer, the samples were ground with 80–2500 grit paper and then polished with some aluminum—oxide paste. The pretreated samples were then placed in a room-temperature phosphate bath for 50 min to remove impurities and then washed with distilled water and alcohol.

We aimed to prepare a thicker and harder oxide coating with a better protective effect than the layers produced with sulfuric acid anodization; therefore, based on the literature, we chose oxalic acid as the electrolyte and galvanostatic deposition because a softer and more flexible layer can be produced potentiostatically [34]. In our previous work, we studied the effect of anodizing conditions on the layer thickness, structure, and hardness of A356 alloy. Then, when planning the research conditions for this research, pointing to tribological properties, we tried to choose the most interesting parameters [33].

The properties of the formed layers can be controlled by the concentration (7 wt%) and temperature of the electrolyte, the applied current density, and the applied time [35,36] (these control parameters and setups can be found in Table 2), so the anodizing equipment had to be designed to allow this set of parameters. Our process was based on the German anodizing standard [37].

**Table 2.** Anodizing parameters (MX—original sample; $MX_{PT}$—the post-treatment samples).

|  | M1 | $M1_{PT}$ | M2 | $M2_{PT}$ | M3 | $M3_{PT}$ | M4 | $M4_{PT}$ | M5 | $M5_{PT}$ |
|---|---|---|---|---|---|---|---|---|---|---|
| j (A/dm$^2$) | 2 |  | 2 |  | 2 |  | 6 |  | 6 |  |
| t (h) | 1 |  | 2 |  | 4 |  | 1 |  | 2 |  |
| Post-treated | - | X | - | X | - | X | - | X | - | X |

The pretreated aluminum workpiece is used as the anode connecting to the positive terminal of a DC power supply, and the aluminum cathode is connected to the negative terminal of the supply. The cathode is rectangle shaped, and the distance of 6 cm between the anode and cathode is the same. The efficiency of the anodization is increased by controlling the temperature of the electrolyte and stirring it (200 revolutions per minute—rpm). Constant temperature, 10 °C, is used. The oxide layer produced by the anodic method is porous, so it does not protect the base metal from corrosion in an aggressive environment. Post-treatment and pore sealing are carried out to increase corrosion resistance or, if required, to decorate. The pores of the oxide layer can be closed using physical (impregnation) or chemical methods. During chemical pore sealing, the oxidized sample is treated in hot water or steam, and then the aluminum oxide turns into monohydrate by absorbing water, and the pores are closed due to the increase in volume [38–40]. We were curious how the post-treatment changes the mechanical properties of the layers, so we made two samples with each parameter and then post-treated (PT) one of them by cooking. (These are the samples $MX_{PT}$, where X = 1, 2, 3, 4, 5).

*2.2. Surface Microstructure and Thickness Measurements*

Microstructures of the base alloy and the anodized layers were studied using digital microscopy (Keyence (Osaka, Japan) VHX-2000), X-ray Diffractometer (XRD, Rigaku Smartlab, Tokyo, Japan) and scanning electron microscopy (SEM, TESCAN VEGA and FEI Quanta 3D FEG + EDAX EBSD).

The anodized sample was cut and embedded in epoxy resin. It was then ground and polished. The layer thickness was measured at 7-7 points on the entire cross-sectional surface on both longitudinal sides, at equal distances from each other, and was averaged.

For the chemical characterization of a sample, Energy-dispersive X-ray spectroscopy (EDS) was used. A ZAF correction was applied to the concentrations. ZAF correction means a correction that considers the following three effects on the characteristic X-ray intensity when performing quantitative analysis: (1) atomic number (Z) effect, (2) absorption (A) effect, and (3) fluorescence excitation (F) effect.

### 2.3. Hardness Measurements

One of the important conditions for the use of aluminum in the automotive industry is adequate hardness. The hardness of the oxide layers is measured by indenting a diamond tip on the cross-sectional microscopic grind of the anodized workpiece (or test specimen). Hardness measurements were performed with an Anton Paar MHT-4 type microhardness tester. The device comes with a Vickers hardness measuring head, which is built into a Leica MEF4M stereomicroscope. Image processing is assisted by the black-and-white CCD camera also built into the stereomicroscope and the black-and-white CRT monitor connected to it. The Vickers hardness value was calculated from the Equation (1). $F$ is the applied load in (N), and $A$ is the surface of the indentation, $d$ is the value of the indentation's diagonal in (mm). 1.854 is the coefficient, which comes from the Vickers indenter's geometry [41]:

$$HV = \frac{F}{A} \cong 1.854 \cdot \left(\frac{F}{d^2}\right) \qquad (1)$$

### 2.4. Tribological Measurements

In recent decades, the wear properties of aluminum-based composites have been investigated in numerous studies. Bhansali and Mehrabian compared the tribological properties of composites reinforced with alumina and SiC [42]. Yang et al. investigated the properties of aluminum reinforced with graphite particles [43]. Prasad et al. reinforced A356.2 alloy with rice husk ash [44], while Abbaipour et al. reinforced A356 alloy with carbon nanotubes [45].

Experimental results have shown that the properties can be improved by reinforcement. However, the literature is sparse on what happens when a wear-resistant layer is made without additives.

For the tribological measurements, the Optimol SRV®5 Test System—which is a ball-on-disc tribometer—was used. The ball-on-disc is a wear test method in which the sliding contact is brought about by pushing a ball specimen onto a rotating disc specimen under a constant load. During the measurements, the samples were placed in a 3D-printed Ø10 cm sample holder suitable for the device's clamping connections. In order to reach an acceptable level of horizontality, the specimens were fixed in the holder with epoxy resin, which filled the gap between the specimen and the holder (Figure 2). After that, a Ø1 cm $ZrO_2$ ball was installed on the machine as a counterpart. To begin with, we performed a couple of test measurements, in which we set and changed the test parameters based on data found in the literature [46–51]. Based on these preliminary measurements, we selected the final measurement data (Table 3). Based on the standard deviation of the wear properties of the test specimens, literature data, and experience from preliminary test measurements, the range of motion of the parameters has narrowed quite a bit.

**Table 3.** Examination parameters.

| | |
|---|---|
| Sliding circle diameter (mm) | 20 |
| Sliding speed (m/s) | 0.2 |
| Applied load (N) | 10 |
| RPM (Rounds per minutes) (1/min) | 191.4 |
| Applied time (min) | 7 |

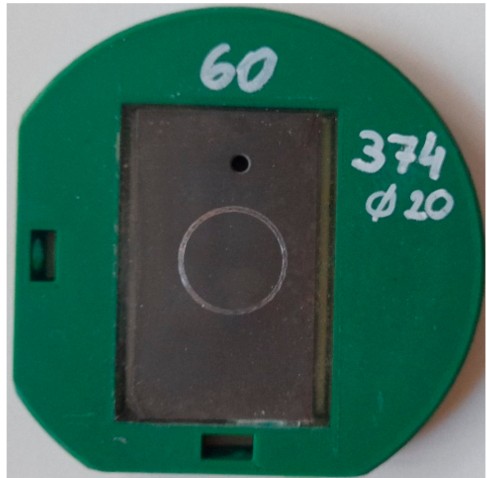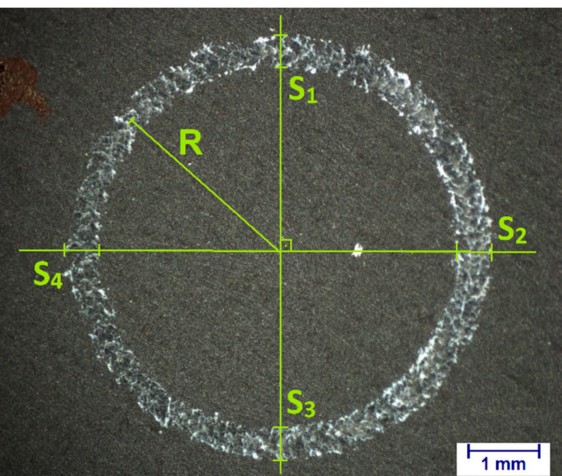

**Figure 2.** Anodized sample (j = 2 A/dm$^2$; t = 1 h) in the holder and resulting wear mark.

After the abrasion test, we determined the nature of the wear using optical and electron microscopy, while the wear volume was based on the width of the wear marks. These parameters were related to the hardness and thickness of the formed layer.

The wear volume was determined based on the ISO20808 standard by measuring the profile cross section (S) at 4 different points of the sample and then the diameter of the circle, from which the radius can be calculated (R) (Figure 2):

$$V = \frac{\pi \cdot R \cdot (S_1 + S_2 + S_3 + S_4)}{2} \tag{2}$$

## 3. Results and Discussion

### 3.1. Microstructure of the Anodized Layers

Figure 3 shows the SEM images on microstructures of the anodized surfaces. The application of low current density j = 2 A/dm$^2$ and t = 4 h time resulted in a relatively compact surface (see Figure 3a). However, when using a higher current density j = 6 A/dm$^2$, an entirely different surface can be observed, like a beehive structure (Figure 3b). With the post-treatment, the pores can be reduced in size and be better closed (Figure 3c,d).

Our previous work [33] showed that the layer thickness does not increase linearly with the oxidation time but reaches a limit, characteristic of the process, so it can be concluded that layer thickness is determined by the quality, concentration, temperature, and current conditions of the electrolyte solution. If the sample reaches the limit value, it does not thicken any further, but a chemical redissolution begins, and in addition to this, the base metal also undergoes deformation.

The layer thickness and the hardness of the samples are in Table 4.

Figure 4 shows the cross section and indentation marks of anodized samples. The indentation marks are clearly visible and show no possible pore breakage, thus the measurements can be considered reliable; however, an instrumented hardness measurement may be necessary for more precise observations.

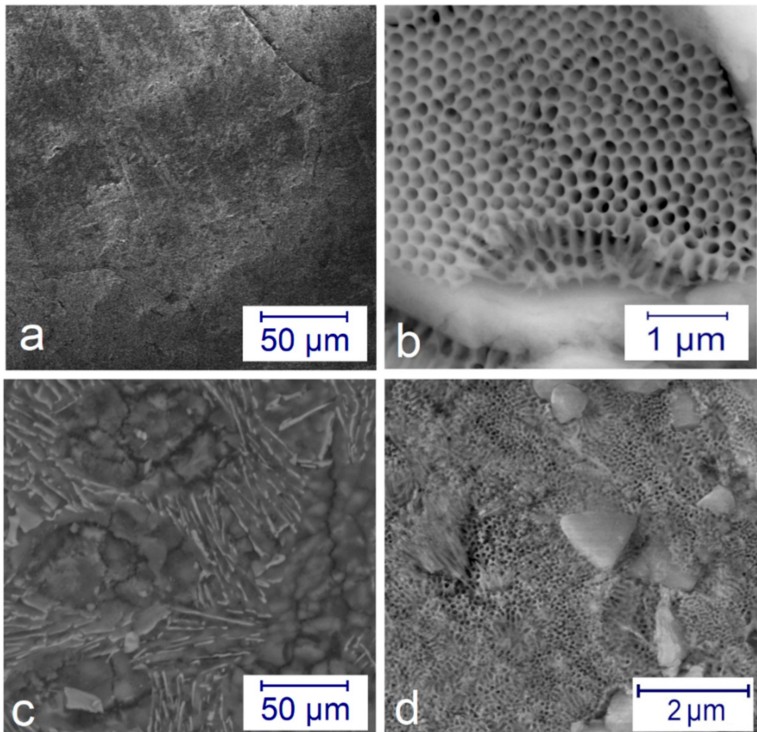

**Figure 3.** SEM pictures on the microstructure of the anodized layers: (**a**) a compact surface at j = 2 A/dm² at 4 h; (**b**) beehive-like structure formed at j = 6 A/dm² for 1 h in different magnifications; (**c**) compact surface at j = 6 A/dm² for 1 h after post-treatment; (**d**) half compact—half porous surface at j = 2 A/dm² for 4 h after post-treatment.

The hardness value of the substrate was 79 ± 1 HV.

Based on previous tests [33], we found that the oxide layer formed on the surface is $\gamma$-$Al_2O_3$, in which there are silicon precipitates.

This was also confirmed by the hardness measurements; we measured a lower hardness in the layer and a higher hardness in the silicon precipitates.

Figure 5 shows the hardness and the layer thickness values of the samples. The low layer thickness samples show higher hardness values.

**Table 4.** Layer thickness and hardness of the samples (MX—original sample; $MX_{PT}$—the post-treatment samples).

|  | M1 | $M1_{PT}$ | M2 | $M2_{PT}$ | M3 | $M3_{PT}$ | M4 | $M4_{PT}$ | M5 | $M5_{PT}$ |
|---|---|---|---|---|---|---|---|---|---|---|
| Layer thickness (µm) | 41.00 | 36.25 | 48.25 | 46.00 | 70.00 | 54.50 | 124.25 | 108.25 | 83.50 | 69.25 |
| Deviation | 5.29 | 2.50 | 1.26 | 2.83 | 1.15 | 3.87 | 11.98 | 2.06 | 1.73 | 3.20 |
| Hardness (HV) | 489 | 426 | 465 | 531 | 536 | 420 | 326 | 341 | 508 | 417 |
| Deviation | 48.9 | 42.6 | 46.5 | 53.1 | 63.6 | 81.9 | 60.0 | 34.1 | 50.8 | 41.7 |

The chemical composition of the samples determined with EDS is shown in Figure 6. The samples were arranged in ascending order according to their layer thickness. Based on the measurement data, we can conclude that the average oxygen content of the samples is 45 ± 1.8 wt% (red), the aluminum content is 38 ± 6.3 wt% (blue), the silicon content is 16 ± 5.6 wt% (green), and the magnesium content is 0.3 ± 0.06 wt% (cyan). Analyzing the obtained concentrations, we could not draw a general conclusion about how much the composition of the layer affects the different mechanical properties.

### 3.2. Tribological Examination

In the ball-on-disc tests, we found that complete wear usually occurs after the first major widening of the frictional force shown in the diagrams. We found that the closeness of the coating wear is indicated by the widening of the measured frictional force (Figure 7), which results from the fact that the ball starts to "bounce" in the wear track. This resulted in the wear track being uneven. As a result, the exact location of wear can only be estimated; it cannot be clearly stated. For the hardest materials, we looked at the time of complete wear, and then all the tests were performed with this time (and thus turning around). Thus, graphs and wear marks can be compared with the same rotation, load, and speed.

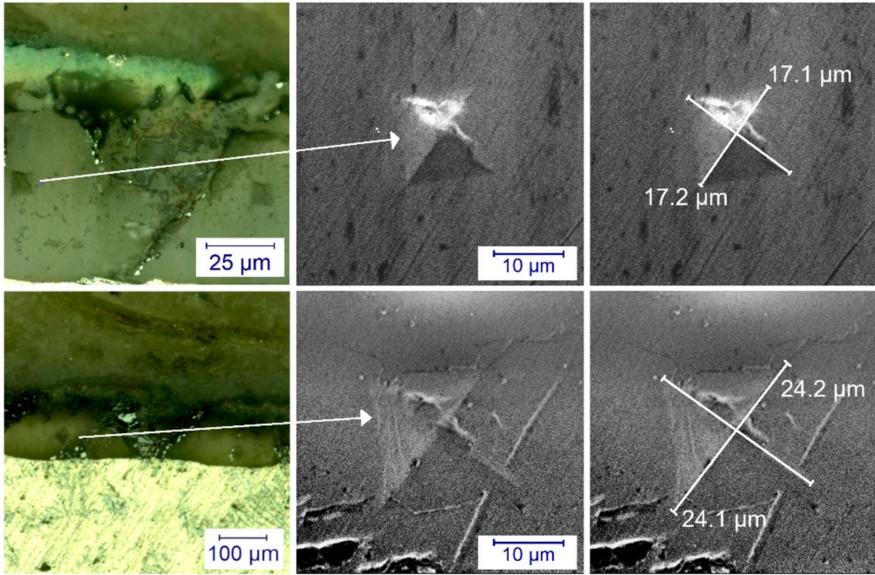

**Figure 4.** Optical microscope and scanning electron microscope image of M3 (**first row**) and M2 (**second row**) samples on the cross section. It can be seen that the diagonals are smaller in the case of the harder sample, while the diagonals of the pattern are larger in the case of the softer sample.

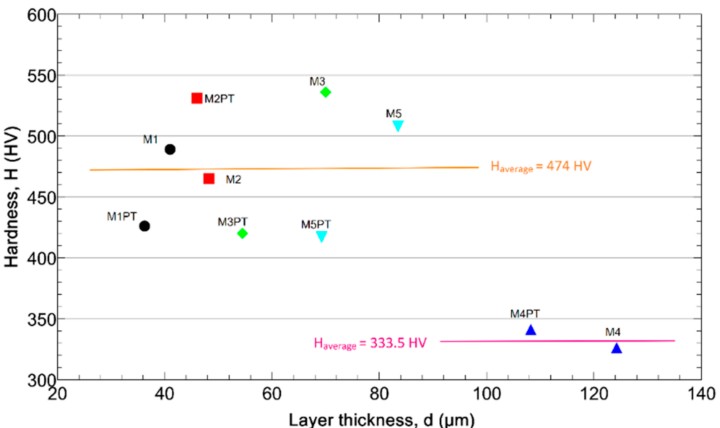

**Figure 5.** The layer hardness as a function of the thickness.

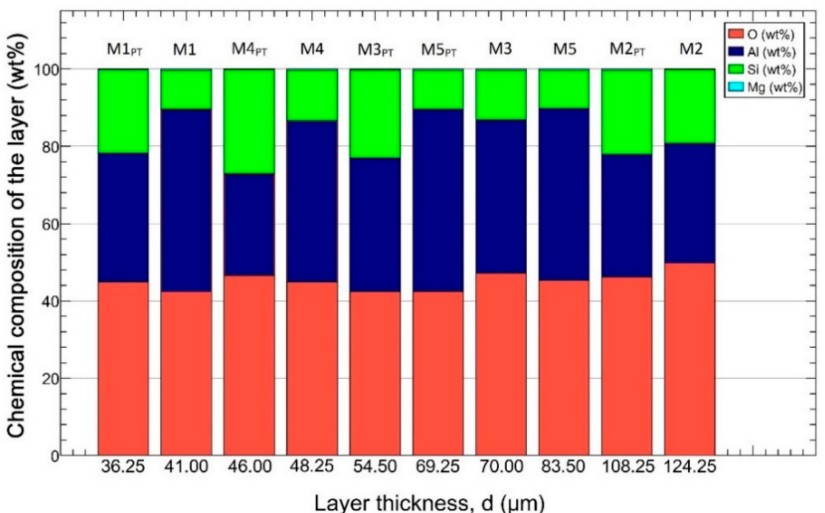

**Figure 6.** The chemical composition of the samples. The samples were arranged by the layer thickness.

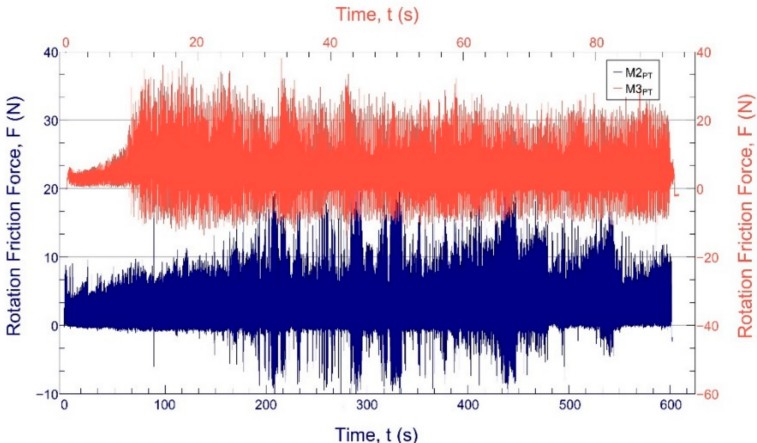

**Figure 7.** The sliding-friction force function of the test time in the case of the sample with the largest wear volume (M3$_{PT}$, orange) and the smallest wear volume (M2$_{PT}$, blue).

First, we examined the surface of the worn samples with optical and electron microscopy (Figures 8 and 9). With the help of the recordings, we examined how much and in what way the sample was worn. The results of this are summarized in Table 5.

Based on this, it can be said that with each measurement, we reached the base metal to a different extent. This phenomenon was investigated with SEM + EDS measurements, as shown in Figure 9.

There were samples that were worn right at the beginning of the test; however, in the case of the M5 sample, the layer was not, or only barely, worn away.

Based on our measurements and the conditions seen after the measurement, the M3 sample values were excluded from the analysis. It can be seen from Figure 10 that while the wear volume decreases with increasing layer thickness, on the contrary, it increases with increasing hardness.

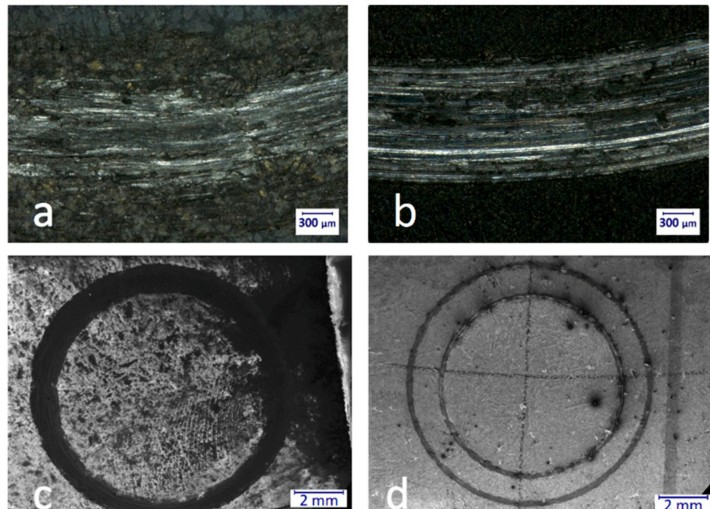

**Figure 8.** Optical microscopic images of a part of the wear track of porous (M5, (**a**)) and nonporous (M4$_{PT}$, (**b**)) samples at 100× magnification (**a**,**b**). In the case of M5, the wear mark is more fragmented and not as continuous as in the case of M4$_{PT}$. The reason for this is that while the hardness of M5 is 508 HV, the hardness of M4$_{PT}$ is "only" 341 HV. A scanning electron microscope was used to evaluate the wear marks more precisely. The SEM images of porous sample (**c**) and the nonporous (**d**) sample at 20× magnification.

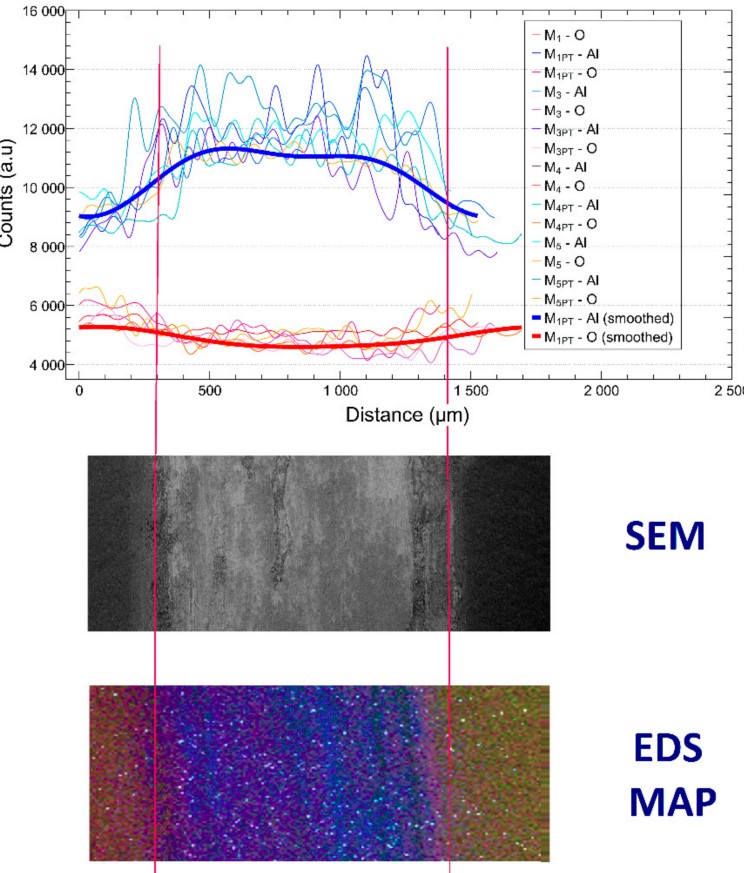

**Figure 9.** EDS signals of the worn samples, SEM image of the wear mark (middle), and the EDS map of the wear mark (bottom, blue—Al, red—O, green—Si, white—Mg).

**Table 5.** Type of wear. The observed wear properties of the samples were marked with blue background.

| | M1 | M1$_{PT}$ | M2 | M2$_{PT}$ | M3 | M3$_{PT}$ | M4 | M4$_{PT}$ | M5 | M5$_{PT}$ |
|---|---|---|---|---|---|---|---|---|---|---|
| **Wear Track** | | | | | | | | | | |
| fragmented | ■ | ■ | ■ | ■ | ■ | | | ■ | ■ | |
| evenly worn | | | | | | | | | | ■ |
| plastically deformed trace | | | | | | ■ | ■ | | | |
| **Debris** | | | | | | | | | | |
| visible | | | | | | ■ | | ■ | ■ | |
| invisible | ■ | ■ | ■ | ■ | ■ | | ■ | | | ■ |
| **Layer status** | | | | | | | | | | |
| partially damaged | | | | | ■ | ■ | | ■ | ■ | |
| completely damaged | ■ | ■ | ■ | ■ | | | ■ | | | ■ |

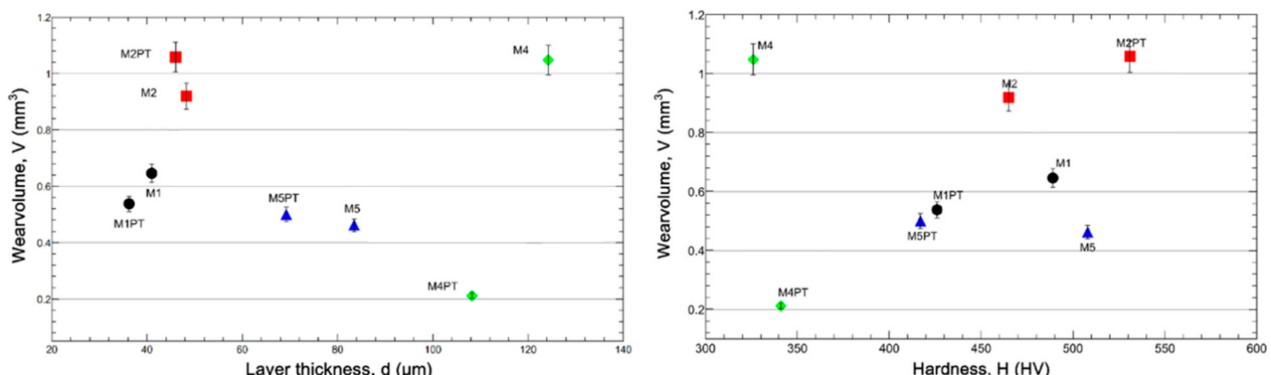

**Figure 10.** Wear volume as a function of layer thickness (**left**) and hardness (**right**).

## 4. Conclusions

Our work focused on the electrochemical improvement of recycled aluminum (A356) alloy anodized at room temperature. The structures of the anodized layers were investigated by using XRD and EBSD measurements, and the efficiency of the anodization was tested by examining the morphology, composition, microhardness, and tribological properties of the formed layers. The main results can be summarized as the following:

(i) It has been shown that the layer formed on the surface of the anodized samples is $\gamma$-$Al_2O_3$, in which there are mainly $SiO_2$ precipitates and a small amount of magnesium. The presence of $SiO_2$ precipitates is an important issue, which opens the door for possible further improvements in the electrochemistry of this alloy system. By fine-tuning the amount of Si in this alloy family, the volume fraction of $SiO_2$ in the $\gamma$-$Al_2O_3$ matrix can also be varied.

(ii) The anodizing process significantly improves the mechanical properties of the sample surface. The hardness of the anodized samples is 5–8 times higher than that of the initial material.

(iii) Experimental results have shown that the structure of the porosity of the anodized surface plays a key role in tribological properties. The wear volume is higher in the case of the porosity having open pores, and the wear volume is lower in the case of closed pores.

**Author Contributions:** A.M. and Á.V.: conceptualization, writing—original draft preparation. A.M.: anodization, cross section analysis, literature review. A.M., Á.V. and D.U.: investigation, data curation. N.Q.C.: formal analysis. M.T.: electrochemical measurement and evaluation. M.W.: sample preparation and microscopy. T.M., S.K., S.S.-P. and K.S.: tribological examination and data curation. Á.V. and N.Q.C.: writing—review and editing, supervision. All authors have read and agreed to the published version of the manuscript.

**Funding:** The research of NQC was supported by the Hungarian Scientific Research Fund OTKA, Grant number K143216. The research of BZN was supported by Project GINOP-2.3.4.-15-2016-00001, which has been implemented with the support provided by European Union and cofinanced by the European Regional Development Fund. Supported by the KDP-2021 Program of the Ministry of Innovation and Technology from the source of the National Research, Development and Innovation Fund.

**Data Availability Statement:** The data that support the findings of this study are all the results of the authors, not available anywhere.

**Conflicts of Interest:** The authors declare that they have no conflict of interest.

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
