# Peer review of "Tribological Examination of Anodized Al-356 for Automotive Use"

_coatings, doi:10.3390/coatings13091642_

Round 1

Reviewer 1 Report

The manuscript is about the tribological properties of anodized Al-356 coatings for potential automotive applications. The primary focus of the research is to investigate how the anodization process affects the friction, wear, and lubrication performance of the coatings, aiming to determine their suitability as protective layers in automotive components.

The manuscript presents results from the experimentation and analyses conducted on anodized Al-356 coatings. The microstructure of the anodized layers is investigated using scanning electron microscopy (SEM). The researchers also studied the influence of anodizing parameters, such as current density and anodization time, on the microstructure and mechanical properties of the coatings.

Furthermore, the manuscript includes a tribological examination, which involves testing the coatings' performance under various conditions using a pin-on-disk apparatus. The friction, wear rates, and wear track characteristics are analyzed to assess the coatings' tribological behavior.

In conclusion, the study emphasizes the importance of optimizing anodizing parameters to achieve desired coating properties. It highlights the potential of anodized Al-356 coatings for automotive applications, particularly in brake disks made from recycled aluminum alloy. The research provides the relationship between anodization conditions, microstructure, hardness, and wear behavior, contributing to the development of wear-resistant and reliable automotive components. However The article looks like a laboratory test report (technical report) and in the way it is written it does not present a technological contribution or to the state of the art related to the anodization of aluminum alloys and its application.

Therefore my recommendation is that the manuscript should not be considered for publication in the "coatings" Journal.

Here my comments:

*Page 1, line 40  What etc means? Please mention all the properties that are not infinite. Be more specific about the properties of aluminum.

*Page 3 line 100 : What means with structural quality of the oxide?   Is it about crystalline structure?

*Page 3, line 101: What does "so" means? "presence of foreign inclusions and so…" I recommend not using words like 'etc' and 'so.' Be specific in the parameters that have effects On the

material properties.

*Page 3 Line 113: Please provide the chemical composition of the electrolyte used. I understand 

that the author used a methodology published before. However, it is helpful to the readers to find these essential parameters in the body of the text.

*Page 3, Lines 138 and 139. How looks the anodized sample (to the naked eye)? Authors must provide an image of the anodized piece that can reveal the aesthetic appearance of the anodizing result.

*What are the measurement conditions of the SEM?

The SEM images are of deficient quality for their magnification ratio. The authors should improve the SEM images and show better quality images (Resolution), allowing us to see the distribution and size of the pores under different anodizing conditions.

*The experimental methods and conditions of measurements are not well described. The current description does not guarantee the reproducibility of the experiments, particularly in the following issues:

*What current source or voltage source did you use for the anodizing process?

*Authors must justify why they use a galvanostatic rather than a potentiostatic regime to fabricate porous aluminium.

*Various articles and previous reports have shown formation rates of the alumina layer more regularly under constant potential conditions. Could the significant variation in the thicknesses obtained be due to this condition?

Please, do a deeper discussion about Al anodizing conditions. The bibliography requires revision, explicitly focusing on the model concerning the development of aluminum oxide in the anodization process. Recent years have seen extensive research in this area, necessitating that the portrayal of oxide layer formation relies on publications from the last  years.

The analysis of the tribology section, which should be the main section of the article, is poor, and has no correlation with the other techniques performed.

Finally, the authors are only testing two anodizing conditions (or currents), which have not been justified in their choice. The graphs of Wear volume and hardness as a function of layer thickness are sparse, and the trend or correlation (which the authors show as a straight line). The measurements carried out are not sufficient to establish this trend and more experimentation and repetitions must be done, as well as a statistical analysis to verify that this result is reproducible.

Author Response

Dear Reviewer!

We are grateful for the Referees detailed response, which can significantly improve the quality of our paper! Please see our reflections in the followings.   

*Page 1, line 40 What etc means? Please mention all the properties that are not infinite. Be more specific about the properties of aluminum.

It has replaced the various types of steels in a wide range of applications due to having a strength to weight ratio superior to steel. Under the influence of oxygen, aluminum forms a natural protective oxide coating, which is why it is highly corrosion resistant. Various surface treatment methods, such as anodizing, painting or varnishing, can further improve this property. It has a low melting point and density. In the molten state, it can be processed in several ways. Furthermore, it can be easily shaped, welded, and as metals usually, it is recyclable, and its quality does not decrease significantly during the process. Remelting aluminum requires lower energy: the energy required to produce the primary metal is only approximately 5% is required in the recycling process.

*Page 3 line 100: What means with structural quality of the oxide?   Is it about crystalline structure? (is now start from line 107)

The anodization of aluminum can be explained according to the Keller-Hunter-Robinson model [26-27], that the liberated oxygen reacts with the aluminum and turns into a dense, homogeneously distributed oxide film, and from this an oxide layer with a thickness dependent on the cell voltage is formed. The oxide film is dielectric in nature, so the current passes through it only as long as the metal ions leaving the base metal are able to diffuse in the oxide-electrolyte boundary phase. The formed aluminum oxide can be dissolved by the electrolyte, then the oxide layer loosens at the defect sites and pores are formed, where the aluminum oxide is re-formed. This process continues until the entire sample is covered with a porous layer, the thickness of which depends on the treatment time. The pore channels penetrate to the oxide barrier layer.

According to the Murphy-Michelson model [28-29], after the formation of the solid oxide barrier layer composed of Al2O3, a part of the aluminum oxide on the outer surface of the layer is transformed into hydroxide by water absorption, and the formation of the barrier layer is continuously renewed on the metal side. According to this theory, pore formation is not a prerequisite for oxide formation. During anodic oxidation, the oxide layer is not detached from the electrolyte onto the surface of the carrier, but is formed through the transformation of the outer layer. It follows that the physical and mechanical properties of the surface metal layer, its chemical composition, tissue structure and crystallographic structure, and its technological background influence the qualitative and structural properties of the formed metal oxide.

Based on the nucleation model [30-31] the oxide formation does not start homogeneously on the metal surface, but tiny solution seeds appear scattered at given points. The process takes place so quickly that a homogeneous, continuous oxide layer can form. These nuclei depend on experimental parameters such as electrolyte concentration and cell voltage.

[26] Keller, F.; Hunter M.S.; Robinson, D.L. Structural features of oxide coatings on aluminum. J. Electrochem. Soc. 1953, 100, 411.

[27] Terashima, A.; Iwai, M.; Kikuchi, T. Nanomorphological changes of anodic aluminum oxide fabricated by anodizing in various phosphate solutions over a wide pH range. App. Surf. Sci. 2022605, 154687.

[28] Csokan, P. The Mechanism of Nucleation and Electrochemical Transport Processes in Oxide Formation During Anodic Oxidation of Aluminium. Trans. of the IMF 1973, 51(1), 6-12.

[29] Pashchanka, M. Conceptual progress for explaining and predicting self-organization on anodized aluminum surfaces. Nanomat., 202111(9), 2271.

[30] Csokan, P. Nucleation mechanism in oxide formation during anodic oxidation of aluminum. Adv. in Corr. Sci. and Techn.,1980,239-356.

[31] Lee, W.; Schwirn, K.; Steinhart, M.; Pippel, E.; Scholz, R.; Gösele, U. Structural engineering of nanoporous anodic aluminium oxide by pulse anodization of aluminium. Nature nanotech. 2008,3(4), 234-239.

The aim of Zhu et al is to determine the influence of the Si content and the morphology of the Si particles on the anodizing reaction. Based on their results, by increasing the Si content, the Al phase appeared to be refined due to more nucleation of eutectic Si, and the fraction of the eutectic phase increased. In the unmodified alloys, the eutectic Si particles showed a polygonal flake shape and formed a continuous branched network [32].

[32] Zhu, B.; Seifeddine, S.; Persson, P. O.; Jarfors, A. E.; Leisner, P.; Zanella, C. A study of formation and growth of the anodised surface layer on cast Al-Si alloys based on different analytical techniques. Mat. & Des., 2008101, 254-262.

This is also confirmed by our own measurement, that after anodization, an amorphous aluminum oxide layer is formed, with Si precipitates in it [33].

[33] Musza, A.; Ugi, D.; Vida, Á.; Chinh, N. Q. Study of Anodic Film’s Surface and Hardness on A356 Aluminum Alloys, Using Scanning Electron Microscope and In-Situ Nanoindentation. Coatings, 2022, 12(10), 1528.

*Page 3, line 101: What does "so" means? "presence of foreign inclusions and so…" I recommend not using words like 'etc' and 'so.' Be specific in the parameters that have effects on the material properties. 

corrected.

*Page 3 Line 113: Please provide the chemical composition of the electrolyte used. I understand that the author used a methodology published before. However, it is helpful to the readers to find these essential parameters in the body of the text. 

corrected - (now line 162)

*Page 3, Lines 138 and 139. How looks the anodized sample (to the naked eye)? Authors must provide an image of the anodized piece that can reveal the aesthetic appearance of the anodizing result.

Please see  Figure 1.

*What are the measurement conditions of the SEM?

The SEM images are of deficient quality for their magnification ratio. The authors should improve the SEM images and show better quality images (Resolution), allowing us to see the distribution and size of the pores under different anodizing conditions.

 corrected.

*The experimental methods and conditions of measurements are not well described. The current description does not guarantee the reproducibility of the experiments, particularly in the following issues: 

 *What current source or voltage source did you use for the anodizing process? corrected

*Authors must justify why they use a galvanostatic rather than a potentiostatic regime to fabricate porous aluminium.

Our aim was to prepare thicker and harder oxide coating with a good protective effect than the layers produced by sulfuric acid anodization, therefore, based on the literature, we chose oxalic acid as the electrolyte and galvanostatic deposition, because a softer and more flexible layer can be produced potentiostatically [34].

[34] Kaltenbrunner, M.; Stadler, P.; Schwödiauer, R.; Hassel, A. W.; Sariciftci, N. S.; Bauer, S. Anodized aluminum oxide thin films for room‐temperature‐processed, flexible, low‐voltage organic non‐volatile memory elements with excellent charge retention. Adv. Mat., 201123(42), 4892-4896.

*Various articles and previous reports have shown formation rates of the alumina layer more regularly under constant potential conditions. Could the significant variation in the thicknesses obtained be due to this condition?

It is known that the layer thickness of the sample depends on the anodizing conditions. In our opinion, the fact that we managed to keep the process at a constant current and at a constant but low temperature, as well as the treatment time was set by monitoring the change in voltage, thus we managed to reduce the potential re-dissolution of the layer, thus we were able to obtain a thicker layer.

Please, do a deeper discussion about Al anodizing conditions. The bibliography requires revision, explicitly focusing on the model concerning the development of aluminum oxide in the anodization process. Recent years have seen extensive research in this area, necessitating that the portrayal of oxide layer formation relies on publications from the last years.

Corrected in previous answer.

The analysis of the tribology section, which should be the main section of the article, is poor, and has no correlation with the other techniques performed.

Finally, the authors are only testing two anodizing conditions (or currents), which have not been justified in their choice.

The treatment methods were selected on the basis of preliminary experiments, with which we managed to produce the best/most interesting, and thus sometimes the most extreme mechanical properties (layer thickness, hardness, porous or non-porous surface).

The graphs of Wear volume and hardness as a function of layer thickness are sparse, and the trend or correlation (which the authors show as a straight line). The measurements carried out are not sufficient to establish this trend and more experimentation and repetitions must be done, as well as a statistical analysis to verify that this result is reproducible.

Thank you for the comment, the lines were removed from graphs to not to give the appearance of a statistical analysis.

Reviewer 2 Report

Dear authors!

Undoubtedly, the research topic is relevant. The study of the mechanical properties of A356 alloy anodizing is very important. But the presented manuscript is sloppy and requires careful revision.

The text is hard to read in places. The presentation needs to be better structured.

1) Some preliminary characterization of the original A356 alloy sample is needed. Is it standard? What is the chemical composition?

2) Line 86 – [−5 °C to +5 °C] – replace the square brackets with round brackets.

3) Line 117 – «substrate was recycled, rectangular shaped aluminum A356 alloy» It is not clear what the «substrate was recycled»? Is the recycled A356 alloy being investigated in the work?

4) Line 127 – The samples were prepared with the parameters of the samples with the best mechanical properties in our previous work [26]. It is not clear. That is, you prepared samples, guided by previously obtained data?

5) Line 135 – «we made two samples with each parameter» The unfortunate term is "parameter". Surface treated in different modes.

Probably, here it is necessary to give a link to Tabl 2. (Respectively, swap Tabl 1 and Tabl 2)

5) Tabl 1 and Tabl 2 are not mentioned in the text of the manuscript.

6) Line 137 – an abbreviation appeared without decoding MXPT. PT is probably “post-treatment” In Table 2 is the same – MXpt?

7) Lines 143–145 «For the thickness measurements the optical microscopy was used: a slice of the anodized sample was cut and embedded in epoxy resin. It was then grinded and polished. The thickness was measured in 7 different points and was averaged» And what are the dimensions of the entire A356 alloy sample under study, the slice dimensions and the distances between these 7 points?

8) What is RPM in in Table 1?

9) Figure 2 is missing.

10) Not enough is written about the measurement of microhardness. What were the dimensions of the print? It is advisable to provide photos of standard prints for different samples. Are there any differences in the nature of the prints for them (cracks, deformation)?

11) The text doesn't mention Figure 6.

Figure 6. EDS map of the wear mark (bottom, blue – Al, red – O, green – Si). Where is Mg? (if it is A356 alloy)

Analysis is missing. Just a statement of the results.

12) «It can be seen from Figure 7 that while the wear volume decreases with increasing layer thickness, and right contrary, it increases with increasing hardness». What are your guesses - why does wear increase with increasing hardness?

13) The text of the manuscript contains the last reference [32], where are the rest [33]-[38]?

14) The list of references is designed carelessly, not in accordance with the template.

Dear authors, you should find out the phase composition of the oxidized layer and discuss its effect on the mechanical properties. This would greatly increase the attractiveness of your work.

It is advisable to check the manuscript with a native speaker.

Author Response

Dear Reviewer!

We are grateful for the Referees detailed response, which can significantly improve the quality of our paper! Please see our reflections in the followings.  

1) Some preliminary characterization of the original A356 alloy sample is needed. Is it standard? What is the chemical composition?

As we see, this alloy is not widely used jet in the technical life, on the other hand it is not so complex hypereutectic composition. We have made a SEM+EDS analysis to show the chemical composition of the alloy.

2) Line 86 – [−5 °C to +5 °C] – replace the square brackets with round brackets.

 corrected – now line 98

3) Line 117 – «substrate was recycled, rectangular shaped aluminum A356 alloy» It is not clear what the «substrate was recycled»? Is the recycled A356 alloy being investigated in the work?

The A356 alloy substrate was prepared from recycled aluminum waste (Figure 1.). The nominal chemical composition of the untreated sample (A356 Al-alloy) is Al-7Si-0.29Mg in wt%.  This was checked also by EDS measurement, the results can be found in in Table 1.)

Table 1. Chemical composition of A356

Al (wt%)

Si (wt%)

Mg (wt%)

Others (wt%)

literary

92

7

0.35

0.65

our sample

91

8

0.3

0.7

To remove the natural oxide layer, the samples were grinded with 80-2500 grit paper and then polished with some aluminum – oxide paste. The pre-treated samples were then placed in a room temperature phosphate bath for 50 minutes to remove impurities, and then washed with distilled water and alcohol.

We aimed to prepare a thicker and harder oxide coating with a good protective effect than the layers produced by sulfuric acid anodization, therefore, based on the literature, we chose oxalic acid as the electrolyte and galvanostatic deposition, because a softer and more flexible layer can be produced potentiostatically [34]. The treatment methods were selected on the basis of preliminary experiments, with which we managed to produce the best/most interesting, and thus sometimes the most extreme mechanical properties (layer thickness, hardness, porous or non-porous surface) [33].

The properties of the formed layers can be controlled by the concentration (7 wt%) and temperature of the electrolyte, the applied current density, and the applied time [35-36] (these control parameters and setups can be found in Table 2.), so the anodizing equipment had to be designed to alloy this set of parameters. Our process was based on the German anodizing standard [37].

Table 2. Anodizing parameters (MX – original sample; MXPT – the post-treatment samples)

M1

M1PT

M2

M2PT

M3

M3PT

M4

M4PT

M5

M5PT

j (A/dm2)

2

2

2

6

6

t (h)

1

2

4

1

2

Post-treated

-

X

-

X

-

X

-

X

-

X

The pre-treated aluminum workpiece is used as the anode connecting to the positive terminal of a DC power supply, and the aluminum cathode is connected to the negative terminal of the supply. The cathode was rectangle-shaped, and the distance of 6 cm between the anode and cathode is the same. The efficiency of the anodization has been increased by controlling the temperature of the electrolyte and stirring it (200 rpm). Constant temperature, 10°C was used. The oxide layer produced by the anodic method is porous, so it does not protect the base metal from corrosion in an aggressive environment. Post-treatment and pore sealing are carried out to increase corrosion resistance or, if required, to decorate. The pores of the oxide layer can be closed using physical (impregnation) or chemical methods. During chemical pore sealing, the oxidized sample is treated in hot water or steam, and then the aluminum oxide turns into monohydrate by absorbing water and the pores are closed due to the increase in volume [38-40]. We were curious how the post-treatment changes the mechanical properties of the layers, so we made two samples with each parameter, and then post-treated one of them by cooking. (These are the samples MXPT, where X=1,2,3,4,5).

[34] Kaltenbrunner, M.; Stadler, P.; Schwödiauer, R.; Hassel, A. W.; Sariciftci, N. S.; Bauer, S. Anodized aluminum oxide thin films for room‐temperature‐processed, flexible, low‐voltage organic non‐volatile memory elements with excellent charge retention. Adv. Mat., 201123(42), 4892-4896.

[33] Musza, A.; Ugi, D.; Vida, Á.; Chinh, N. Q. Study of Anodic Film’s Surface and Hardness on A356 Aluminum Alloys, Using Scanning Electron Microscope and In-Situ Nanoindentation. Coatings, 2022, 12(10), 1528.

4) Line 127 – The samples were prepared with the parameters of the samples with the best mechanical properties in our previous work [26]. It is not clear. That is, you prepared samples, guided by previously obtained data?

We apologize for the simplified wording, but as the reviewer thinks, we already have/had preliminary experiments, from which we also wrote an article and we chose for this article from the samples that we found interesting.

In our previous work, we have studied the effect of anodizing conditions on the layer thickness, structure and hardness of A356 alloy. Then, when planning the research conditions for this research, pointing to tribological properties, we tried to choose the most interesting parameters.

5) Line 135 – «we made two samples with each parameter» The unfortunate term is "parameter". Surface treated in different modes.

corrected.

Probably, here it is necessary to give a link to Tabl3 2. (Respectively, swap Tabl 1 and Tabl 2)

corrected.

5) Tabl 1 and Tabl 2 are not mentioned in the text of the manuscript.

corrected.

6) Line 137 – an abbreviation appeared without decoding MXPT. PT is probably “post-treatment”In Table 2 is the same – MXpt?

We apologize for the simplified wording. The samples on which post-treatment was performed were marked with MXPT, where X=1,2,3,4,5. We expanded this definition in line 158.

7) Lines 143–145 «For the thickness measurements the optical microscopy was used: a slice of the anodized sample was cut and embedded in epoxy resin. It was then grinded and polished. The thickness was measured in 7 different points and was averaged» And what are the dimensions of the entire A356 alloy sample under study, the slice dimensions and the distances between these 7 points?

corrected.

8) What is RPM in in Table 1?

RPM is the rotation speed of the ball (rotation per minute).

9) Figure 2 is missing.

Corrected.

10) Not enough is written about the measurement of microhardness. What were the dimensions of the print? It is advisable to provide photos of standard prints for different samples. Are there any differences in the nature of the prints for them (cracks, deformation)?

Based on previous tests, we see that the formed oxide layer is an amorphous aluminum oxide matrix, in which there are silicon precipitates.

This was also confirmed by the hardness measurements since we measured a lower hardness in the amorphous layer and a higher hardness in the silicon precipitate. That's why we tried to perform as many measurements as possible, at equal distances from each other.

11) The text doesn't mention Figure 6.

corrected.

Figure 6. EDS map of the wear mark (bottom, blue – Al, red – O, green – Si). Where is Mg? (if it is A356 alloy)

corrected.

Analysis is missing. Just a statement of the results.

The mapping was only used to see, what extent the base metal appears in the wear mark. With this information we were able to prepare Table 5.

12) «It can be seen from Figure 7 that while the wear volume decreases with increasing layer thickness, and right contrary, it increases with increasing hardness». What are your guesses - why does wear increase with increasing hardness?

This is in important question; however, we have no exact answers yet. Surface roughness (and by this the heat accumulated during tribology measurements), precipitations in the layer and maybe other phenomena may play key role. Our research team continues this work to find the answer to this question.

13) The text of the manuscript contains the last reference [32], where are the rest [33]-[38]?

References 33-38 (now is 45-50) are the literature we used to determine the parameters of tribological wear. We replaced the indication of this in line 225.

14) The list of references is designed carelessly, not in accordance with the template

Corrected.

Dear authors, you should find out the phase composition of the oxidized layer and discuss its effect on the mechanical properties. This would greatly increase the attractiveness of your work.

Thank you for this notice, we totally agree. However, the conducted EDS measurements (now in Figure 6.) show no clear connection between composition and general properties of the layers.

Round 2

Reviewer 1 Report

The manuscript has been consistently improved.

However, a physicochemical analysis of the results is still lacking. The results continue to appear scattered and uncorrelated. The conclusions are not useful and only summarize the results obtained. I invite the authors to reconsider the writing of the conclusions, including a more proactive and useful analysis for the readers.

The article can be published once the conclusions section is considerably improved.

Author Response

Dear Reviewer,

thank you for the valuable feedback and support. Based on Your response, the conclusions part was cleaned and focused.

We hope this improvement can be suitable for publication in your point of view.

Reviewer 2 Report

Dear authors,

I am completely satisfied with your answers and correction of the manuscript.

Now I have a few minor remarks and one more significant one. 

1) In Table 3 - "RPM (Rotation per minute)".  Probably, it's "Revolutions Per Minutes" or “Rounds Per Minutes”?

Please complete Table 3 in the same way as Table 4: quantity (dimension) - value. 

2) Lines 124-127 - it is not clear what influences what? And incomprehensible terminology. “Тissue structure” is morphology of coating? By “crystallographic structure” is meant the “phase composition”? 

3) The conclusion contains the amazing phrase: «Through structural tests, we established that our layer formed on the surface of the initial material is a γ-Al2O3 matrix, in which there are mainly SiO2 precipitates and a small amount of magnesium». What are these “structural tests”? X-ray diffraction? Why is there no mention of this in the main text? If these are EDS results, then these are not “structural tests” at all, but the determination of the elemental composition. How can one judge on their basis that the matrix is γ-Al2O3? Maybe this is another modification of Al2O3? 

4) It is not good to start the "Conclusion" with the words "The purpose of our article ..." Write better "This article showed how you can improve the mechanical and tribological properties ..."

5) I do not speak English well enough to judge the quality of the presentation. But still, it seems to me that many proposals need to be edited. Therefore, I ask you to check the manuscript again with a native speaker or a qualified translator.

Author Response

Dear Reviewer, thank you for your feedback and instructions. Please find our answers in the followings.

1) In Table 3 - "RPM (Rotation per minute)".  Probably, it's "Revolutions Per Minutes" or “Rounds Per Minutes”?

We used Rotation, because it was also mentioned like this in the parameters of the machine, on  the other hand, we agree that Rounds is better, thus we changed this word.

Please complete Table 3 in the same way as Table 4: quantity (dimension) - value. 

Thank you, corrected.

2) Lines 124-127 - it is not clear what influences what? And incomprehensible terminology. “Тissue structure” is morphology of coating? By “crystallographic structure” is meant the “phase composition”? 

Thank you, the sentence is simplified.

3) The conclusion contains the amazing phrase: «Through structural tests, we established that our layer formed on the surface of the initial material is a γ-Al2O3 matrix, in which there are mainly SiO2 precipitates and a small amount of magnesium». What are these “structural tests”? X-ray diffraction? Why is there no mention of this in the main text? If these are EDS results, then these are not “structural tests” at all, but the determination of the elemental composition. How can one judge on their basis that the matrix is γ-Al2O3? Maybe this is another modification of Al2O3? 

We carried out a general structural analysis using X-ray diffraction and Electron Backscattered Diffraction method to obtain the phase structure of the layer. This showed gamma alumina in the highest concentration. The Materials and Methods paragraph was improved like this: Microstructures of the base alloy and the anodized layers were studied by digital mi-croscopy (Keyence VHX-2000), X-Ray Diffractometer (Rigaku Smartlab) and scanning electron microscopy (TESCAN VEGA and FEI Quanta 3D FEG + EDAX EBSD).

4) It is not good to start the "Conclusion" with the words "The purpose of our article ..." Write better "This article showed how you can improve the mechanical and tribological properties ..."

The whole conclusion was changed, thank you for the valuable support.

5) I do not speak English well enough to judge the quality of the presentation. But still, it seems to me that many proposals need to be edited. Therefore, I ask you to check the manuscript again with a native speaker or a qualified translator.

Thank you for the support, the article was double checked by a native speaker.